# Expansion microscopy provides new insights into the cytoskeleton of malaria parasites including the conservation of a conoid

Eloïse Bertiaux[1], Aurélia C. Balestra[2], Lorène Bournonville[1], Vincent Louvel[1], Bohumil Maco[2], Dominique Soldati-Favre[2], Mathieu Brochet[2]*, Paul Guichard[1]*, Virginie Hamel[1]*

1 University of Geneva, Department of Cell Biology, Faculty of Science, Geneva, Switzerland, 2 University of Geneva, Department of Microbiology and Molecular Medicine, Faculty of Medicine, Geneva, Switzerland

⊘ These authors contributed equally to this work.
* Mathieu.Brochet@unige.ch (MB); Paul.Guichard@unige.ch (PG); Virginie.Hamel@unige.ch (VH)

**Data Availability Statement:** All relevant data are within the paper and its Supporting Information files.

## Abstract

Malaria is caused by unicellular *Plasmodium* parasites. *Plasmodium* relies on diverse microtubule cytoskeletal structures for its reproduction, multiplication, and dissemination. Due to the small size of this parasite, its cytoskeleton has been primarily observable by electron microscopy (EM). Here, we demonstrate that the nanoscale cytoskeleton organisation is within reach using ultrastructure expansion microscopy (U-ExM). In developing microgametocytes, U-ExM allows monitoring the dynamic assembly of axonemes and concomitant tubulin polyglutamylation in whole cells. In the invasive merozoite and ookinete forms, U-ExM unveils the diversity across *Plasmodium* stages and species of the subpellicular microtubule arrays that confer cell rigidity. In ookinetes, we additionally identify an apical tubulin ring (ATR) that colocalises with markers of the conoid in related apicomplexan parasites. This tubulin-containing structure was presumed to be lost in *Plasmodium* despite its crucial role in motility and invasion in other apicomplexans. Here, U-ExM reveals that a divergent and considerably reduced form of the conoid is actually conserved in *Plasmodium* species.

## Introduction

Malaria is caused by unicellular *Plasmodium* parasites that belong to the phylum Apicomplexa. More than 6,000 species have been described in this phylum [1] including numerous human and animal pathogens such as *Toxoplasma gondii* and *Cryptosporidia*. During their life cycle, apicomplexan parasites undergo multiple differentiation steps into morphologically distinct forms sustaining either (i) sexual reproduction; (ii) asexual replication; or (iii) dissemination via egress and invasion of host cells [2]. Each of the stages relies on diverse microtubule cytoskeletal structures that either share general characteristics with those of other eukaryotic organisms or are unique to these parasites.

A unique cytoskeletal feature that distinguishes Apicomplexa from other eukaryotes is the apical complex. This complex is assembled in polarised invasive stages, called zoites. It includes

**Funding:** This work was supported by the Swiss National Science Foundation (SNSF) PP00P3_187198 attributed to PG, BSSGI0_155852 and 31003A_179321 to MB, and by the European Research Council ERC ACCENT StG 715289 attributed to PG. MB is an INSERM investigator. MB and PG are EMBO young investigators. EB is supported by an EMBO long-term fellowship (ALTF-284-2019). BM is funded by the European Research Council (ERC) under the European Union's Horizon 2020 research and innovation program under Grant agreement no. 695596. VH is funded by the European Research Council ERC ACCENT StG 715289 and PG by the Swiss National Science Foundation (SNSF) PP00P3_187198. The funders had no role in study design, data collection and analysis, decision to publish, or preparation of the manuscript.

**Competing interests:** The authors have declared that no competing interests exist.

**Abbreviations:** APR, apical polar ring; ATR, apical tubulin ring; EM, electron microscopy; FCS, fetal calf serum; GCβ, guanylyl cyclase beta; GFP, green fluorescent protein; IFA, immunofluorescence assay; IMC, inner membrane complex; MTOC, microtubule organising centre; NHS, N-hydroxysuccinimide; PKG, protein kinase G; ROI, region of interest; RT, room temperature; SAS6L, SAS-6-like; SIM, structured illumination microscopy; STED, stimulated emission depletion; TEM, transmission electron microscopy; U-ExM, ultrastructure expansion microscopy.

specialised secretory organelles and a microtubule organising centre (MTOC) called the apical polar ring (APR). The secretory organelles release various factors necessary for egress, motility, and invasion [3]. An array of subpellicular microtubules nucleates from the APR and confers the shape and stability to the parasite [3]. Two additional electron dense rings have been described above the large and thick APR in multiple apicomplexan parasites, including *Plasmodium* zoites [4,5]. The fine structure and molecular composition of the apical complex differ among apicomplexan parasites and probably reflect different mechanical or functional requirements linked to the range of invaded host cells. For example, the presence or absence of a hollow tapered barrel structure composed of tubulin called the conoid [6] has defined 2 classes within the Apicomplexa. The Conoidasida, such as coccidia, including *T. gondii* and gregarines, possess a conoid [7]. Based on ultrastructural data, *Plasmodium* has been traditionally considered to lack a conoid and belongs to the Aconoidasida [8]. While the structure of the apical complex is relatively well characterised in some apicomplexan parasites such as *T. gondii* [9], its molecular composition remains elusive and difficult to visualise in *Plasmodium*.

Our understanding of *Plasmodium* microtubule-based structures heavily relies on electron microscopy (EM). Fluorescent light microscopy is instrumental to complement EM with the possibility of using multiple markers to infer the dynamic and the molecular composition of *Plasmodium* cytoskeletons. However, owing to the small size of the parasites, subcellular imaging by fluorescent light microscopy still poses a major challenge in *Plasmodium*. Super-resolution techniques have recently been implemented, but structured illumination microscopy (SIM) provides only slightly improved resolution (approximately 120 nm) in comparison to diffraction-limited light microscopy (approximately 240 nm), while stimulated emission depletion (STED) microscopy can reach a resolution of up to 35 nm [10,11]. However, the iron-rich hemozoin crystals present in the asexually replicating blood stages and sexual stages cause cell disintegration when illuminated with the high-power STED laser [12,13]. The implementation of guided or rescue STED has partly circumvented this issue by automatically deactivating the STED depletion laser in highly reflective regions of the sample to prevent local damage [12,13]. Because of the limit of resolution of fluorescent microscopy or the technical limitations of super-resolution microscopy, the structure and molecular composition of the *Plasmodium* microtubule cytoskeleton remain difficult to interrogate.

Here, we reasoned that physical expansion of *Plasmodium* cells in an isotropic fashion by ultrastructure expansion microscopy (U-ExM) [14] could provide an accessible bridge between traditional fluorescent microscopy and EM, as previously shown in *T. gondii* [15,16]. We expanded multiple *Plasmodium* stages and species including 2 zoite stages, *Plasmodium falciparum* schizonts, *Plasmodium berghei* schizonts, and *P. berghei* ookinetes, as well as developing *P. berghei* microgametes. We show that U-ExM resolves the structure of the axonemes, the mitotic hemispindles as well as the subpellicular microtubules. Importantly, it reveals the existence of an apical tubulin ring (ATR) colocalising with markers of the conoid in other parasites that is atop the APR of motile ookinetes, suggesting that the remnant of a conoid was retained in this genus. In conclusion, U-ExM enables visualisation of cytoskeletal structures at a nanoscale resolution in *Plasmodium* and represents an accessible method without the need of specialised microscopes.

## Results

### U-ExM is effective in diverse stages of the *Plasmodium* life cycle

To test the potential of U-ExM [14] to image *Plasmodium*, we compared protocols for fluorescent microscopy of tubulin structures with or without expansion on multiple life cycle stages (**Fig 1**). Briefly, *Plasmodium* parasites were first sedimented on poly-D-lysine coated coverslips

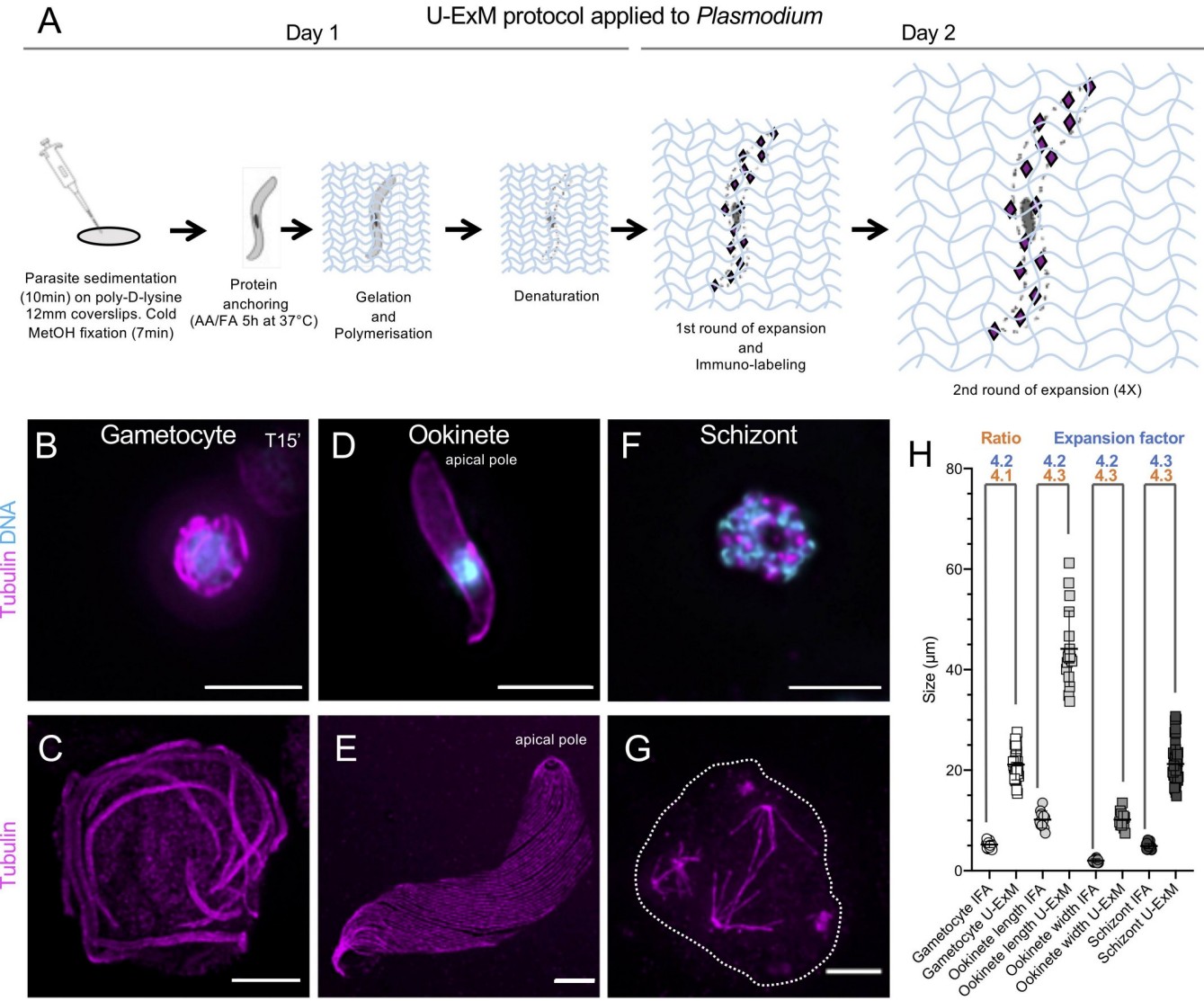

**Fig 1. U-ExM applied to *Plasmodium*.** (A) Schematic illustration of U-ExM protocol applied to *Plasmodium* samples. (B, D, F) Epifluorescence images of a *P. berghei* gametocyte (B), a *P. berghei* ookinete (D), and a *P. falciparum* schizont (F) stained for α- and β-tubulin (magenta, Alexa 568) and DNA (cyan). Scale bar: 5 μm. (C, E, G) Same stages as above expanded using U-ExM and stained for α/β-tubulin (magenta, Alexa 568). Scale bar: 5 μm. (H) Average size of the different studied stages before (IFA) and after expansion (U-ExM). Numbers of Gametocytes: IFA, 25; U-ExM, 28; Ookinetes (length): IFA, 11; U-ExM, 18; Ookinetes (width): IFA, 11; U-ExM, 17; Schizonts: IFA, 30; U-ExM, 47. Orange: ratio between the size before and after expansion are indicated. Blue: gel expansion factor. The raw data can be found in S1 Data. IFA, immunofluorescence assay; U-ExM, ultrastructure expansion microscopy.

prior to cold methanol fixation. Next, proteins of the samples were anchored to a swellable polymer, followed by denaturation and expansion. Immunostaining was subsequently performed post-expansion as previously described [14] (**Fig 1A**).

We first imaged the development of *P. berghei* microgametocytes into microgametes, a process that is essential for sexual reproduction and establishment of infection in the mosquito (**Fig 1B**). During microgametogenesis, the component parts of 8 axonemes are assembled within the gametocyte cytoplasm to form flagellated male gametes in a process called exflagellation. Confocal imaging of cells stained for α/β-tubulin showed a bundle of undistinguishable axonemes coiled around the octoploid nucleus. In contrast, expanded gametocytes resolved individual axonemes (**Fig 1C**), featuring the gain of resolution obtained using U-ExM.

Importantly, comparison of the average diameters of cells with or without expansion indicated an approximately 4.2-fold size increase in the linear dimension, from around 5 to 20 μm (**Fig 1H**, **S1 Fig**).

We then applied U-ExM to *P. berghei* ookinete, a large motile extracellular zoite of 10 to 12 μm in length, responsible for colonisation of the mosquito midgut that displays a subpellicular microtubule network radiating from the APR. Non-expanded ookinetes stained for α/β-tubulin showed a diffuse cytoplasmic signal with an enrichment at the cell periphery and no visible microtubule structures (**Fig 1D**). Expanded cells showed an approximately 4.2-fold increase in both width and length, highlighting the isotropic expansion of the specimen (**Fig 1E and 1H**, **S1 Fig**). This resulted in the clear resolution of individual subpellicular microtubules that radiate from the APR (**Fig 1E**, **S1 Fig**).

Next, we stained schizonts of *P. falciparum*. During erythrocytic schizogony, the parasite undergoes multiple rounds of replications to form up to 32 merozoites per schizont. Merozoites are approximately 1 to 2 μm long zoites that upon rupture of the host erythrocyte quickly reinvade erythrocytes. Confocal imaging of late stage schizonts prior to expansion showed a punctuate signal for α/β-tubulin (**Fig 1F**). Sample expansion showed a 4.3-fold increase in global size without apparent major morphological distortions and highlighted the previously described mitotic hemispindles [13] (**Fig 1G and 1H**, **S1 Fig**).

Altogether, expansion of 3 stages from 2 *Plasmodium* species indicates a consistent approximately 4.2-fold size increase in the linear dimension (**Fig 1H**) and demonstrates that U-ExM can be readily applied to multiple *Plasmodium* stages, revealing subtle details of its internal cytoskeleton organisation.

## U-ExM allows visualisation of axonemal microtubules and their polyglutamylation in fully reconstructed gametocytes

*Plasmodium* assembles a flagellum exclusively prior to the formation of the microgamete in the midgut of the mosquito vector [17]. Microgametogenesis is an explosive development with 3 rounds of genome replication and endomitosis, paralleled by the assembly of 8 axonemes to form 8 microgametes within 10 to 20 minutes of activation [18,19]. The axonemes display the classical organisation with 9 doublets of microtubules arranged in a circular pattern surrounding a central pair of singlet microtubules [20]. Flagellum formation in *Plasmodium* is remarkably different from known models [21]. Axonemes are assembled within the cytoplasm of the microgametocyte independently of intraflagellar transport [22], and the flagellum is formed later, after the initiation of axoneme beating [23]. EM observations indicate that this atypical and rapid synthesis is associated with frequent axoneme misassembly [23]. As the exact mode of assembly remains unknown, an understanding as to the specific origins of these presumed errors might inform us of the molecular mechanisms of axoneme pattern determination [23].

Therefore, we undertook the analysis of axonemal assembly in cellulo in microgametocytes using U-ExM (**Fig 2A–2D**). To do so, we immunolabelled for α/β tubulin microgametocytes that were fixed 15 minutes after gametogenesis activation, when the first exflagellation events are observed. This revealed snapshots of axonemal assembly that have only been visible by EM observations of serial sections. We observed a wide range of patterns, ranging from incomplete axoneme formation where fan-shaped arrays of short microtubules radiate from the basal body (**Fig 2A**, **S2 Fig**) to apparently fully assembled axonemes (**Fig 2C**). A detailed analysis of 25 whole-cell projections revealed that only 4% (1/25) of 15-minute activated gametocytes showed fully formed axonemes, while the others cells presented 3 main and nonexclusive intermediate states. First, 80% of cells (20/25) showed less than 8 axonemes, which could reflect defects in basal body replication or other early impairment of microtubule

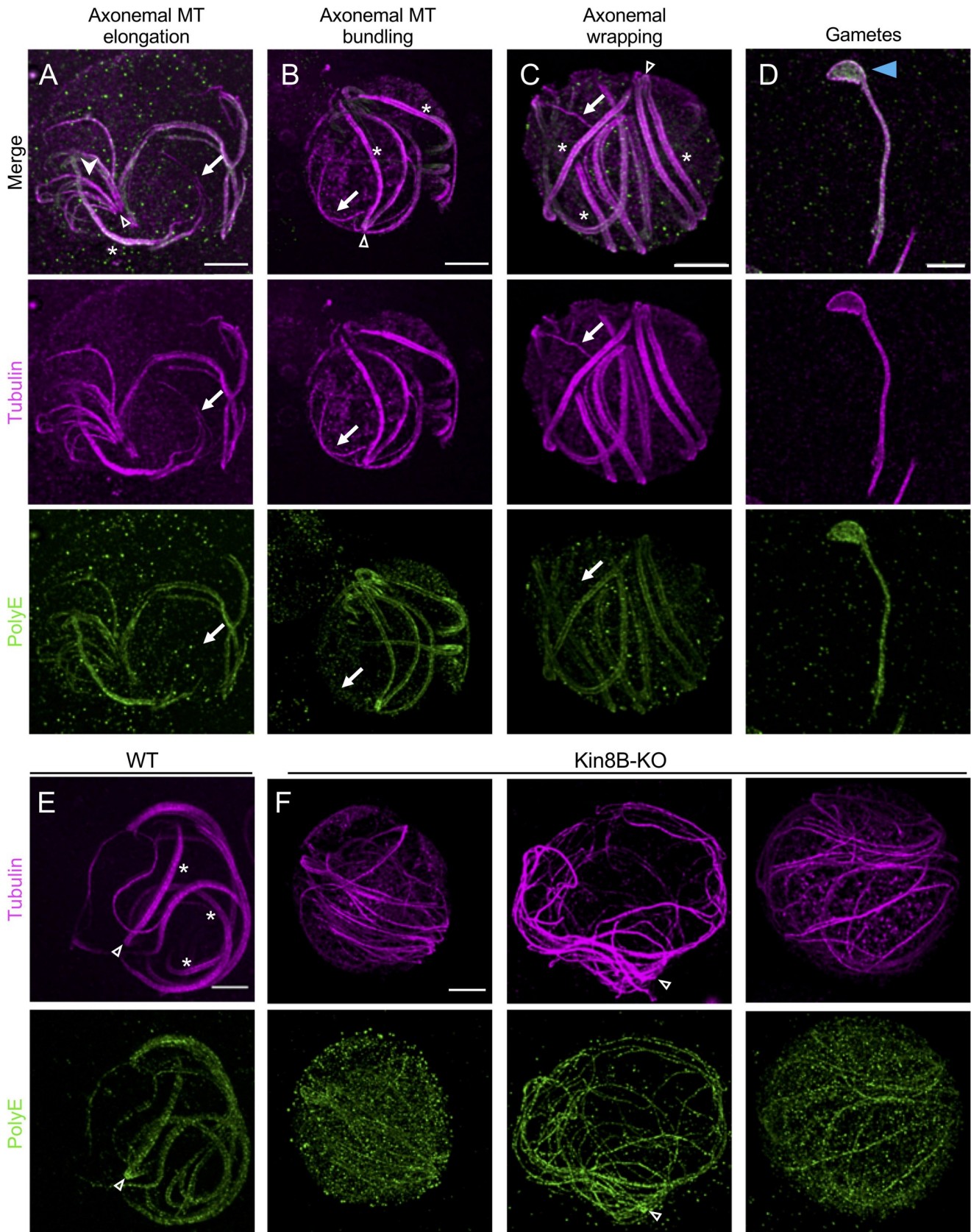

**Fig 2. Axonemes formation in microgametocytes visualised by U-ExM. (A–D)** Gallery of representative confocal images in microgametocytes and microgametes 15 minutes post-activation based on morphological features. Gametocytes were expanded and stained for α- and β-tubulin (magenta, Alexa 568) and PolyE (green, Alexa 488). Arrows show apparent differences between tubulin and PolyE staining. White arrowhead indicates the fan-shaped arrays of microtubules. Blue arrowhead points to the remnant body. Open arrowhead indicates a clearly identifiable basal body position. White asterisk denotes fully formed axonemes. Scale bar: 5 µm. **(E, F)** Expanded WT (E) and Kin8B-KO (F) gametocytes stained for α/β-tubulin (magenta, Alexa 568) and PolyE (green, Alexa 488). Complete axonemes are visible 15 minutes after activation for the WT, while individual unassembled singlets or doublets microtubules are seen in the mutant. Open arrowhead indicates a clearly identifiable basal body position. Scale bar: 5 µm. MT, microtubule; U-ExM, ultrastructure expansion microscopy; WT, wild-type.

polymerisation (**Fig 2A and 2B**, **S2 Fig**). Second, we observed that within single gametocytes, both incomplete and fully assembled axonemes coexist within a shared cytoplasm (**Fig 2B, S2 Fig**). Third, 44% (11/25) of the imaged axonemes displayed apparently free microtubule singlets or doublets (**Fig 2A–2C**, white arrows; **S2 Fig**). In contrast to the frequent observation of misassembled axonemes in developing gametocytes, the axoneme of free microgametes did not show such apparent defects (**Fig 2D**). To ascertain that the observed structures correspond to non-assembled axonemes, we imaged Kin8B-KO microgametocytes where full-length singlet or doublet microtubules are formed but not assembled into the 9+2 pattern [24,25]. As previously reported, we could not observe assembled axonemes in Kin8B-KO microgametocytes (0%; 27/27) 15 minutes after activation (**Fig 2E and 2F**), demonstrating that U-ExM allows visualisation of microtubule singlets or doublets not incorporated into the 9+2 pattern.

To take advantage of the versatility of U-ExM, we then stained gametocytes for polyglutamylated tubulin a posttranslational modification that stabilises microtubules with various roles in regulating flagellar motility [26]. Polyglutamylation (PolyE) was observed on both assembling and full-length axonemes (**Fig 2A–2D**), although we also observed that some microtubules that were not incorporated into mature flagella were lacking PolyE (**Fig 2A–2C**, white arrows; **S2** and **S3 Figs**). These results suggest that PolyE is added onto assembling axonemes and is more strongly detected in fully formed axonemes (**S3 Fig**). To test whether PolyE is dependent on axonemal formation, we specifically stained Kin8B-KO microgametocytes and detected a signal of PolyE on microtubules from non-assembled axonemes as well as a high background in the cytoplasm of the gametocytes, possibly reflecting unincorporated polyglutamylated tubulin dimers or cross-reactivity with other glutamylated proteins whose distribution is altered in the mutant. Consistent with this, polyglutamylated tubulin was not affected in the mutant line (**S3 Fig**). Altogether, these results suggest that PolyE deposition occurs in a timely manner during axoneme formation, but that is independent from complete assembly (**Fig 2E**).

Altogether, U-ExM unveils details of axoneme formation in whole cells to levels that have only been accessible by EM. Based on these results, we anticipate that expansion microscopy will reveal new aspects of *Plasmodium* gametogenesis that have been overlooked by conventional fluorescence microscopy.

## U-ExM resolves the hemispindles and reveals a distinct organisation of subpellicular microtubules in *P. falciparum* and *P. berghei* schizonts

We then analysed expanded 40 to 48 hours *P. falciparum* schizonts stained for α/β tubulin (**Fig 3**). Two distinct microtubule structures were observed in this stage by U-ExM: the mitotic hemispindles and the subpellicular microtubules. In late schizonts, we observed the hemispindles that form an array of microtubules, which radiate from single MTOCs, as revealed by centrin staining (**Fig 3A and 3B** arrowheads indicate centrin dots at the mitotic spindle pole). Per centriolar plaque, we observed an average of 4 hemispindles ranging from 0.5 to 2 µm (**Fig 3C and 3F**), as previously described [27]. Interestingly, the hemispindles were not polyglutamylated as opposed to mitotic spindles in other eukaryotic cells [28], possibly suggesting a requirement for dynamic spindles in *Plasmodium* (**Fig 3D and 3E**).

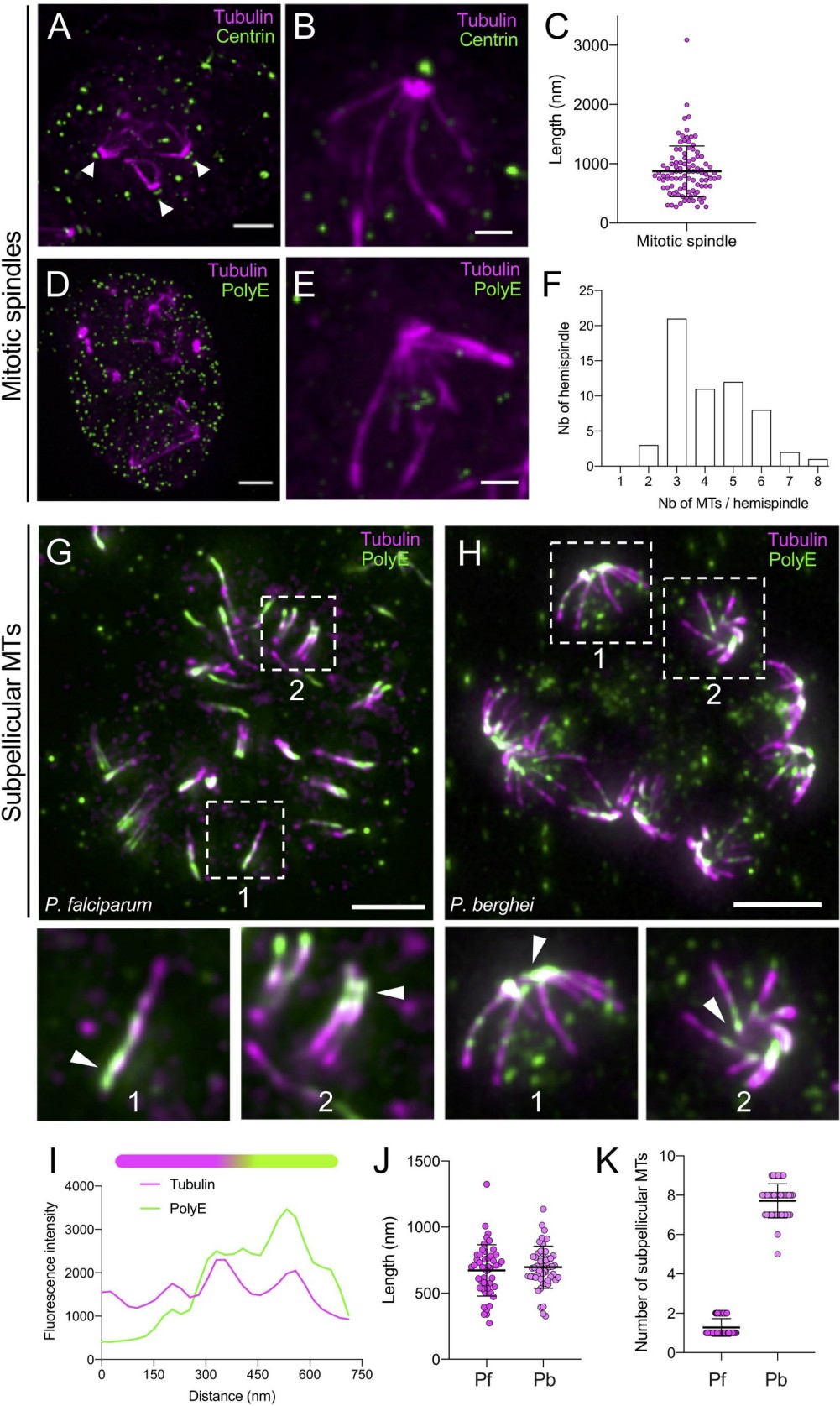

**Fig 3. U-ExM resolves mitotic hemispindles and subpellicular microtubules in *P. falciparum* schizonts.** (**A**) Epifluorescence image of an expanded schizont presenting mitotic spindles stained for α/β-tubulin (magenta, Alexa 568) and centrin (green, Alexa 488). White arrowheads point to centrin staining at spindle poles. Note that some extra dots are visible in the centrin staining. Scale bar: 5 μm. (**B**) Zoom in of an hemispindle stained for α/β-tubulin (magenta, Alexa 568) and centrin (green, Alexa 488). Scale bar: 2 μm. (**C**) Dot plot representing the mitotic spindle length. Average length +/− standard deviation: 873 nm +/− 427, $n = 101$. (**D**) Epifluorescence image of an expanded schizont presenting mitotic spindles stained for α/β-tubulin (magenta, Alexa 568) and PolyE (green, Alexa 488). Scale bar: 5 μm. Note that the mitotic spindle is not polyglutamylated. (**E**) Zoom in of an hemispindle stained for α/β-tubulin (magenta, Alexa 568) and PolyE (green, Alexa 488). Scale bar: 2 μm. (**F**) Histogram representing the number of hemispindles displaying 1, 2, 3, 4, 5, 6, 7, or 8 tubulin structures, $n = 58$. (**G**) Representative epifluorescence images of expanded *P. falciparum* schizonts presenting subpellicular microtubules stained for α/β tubulin (magenta, Alexa 568) and PolyE (green, Alexa 488). Note that the subpellicular microtubules are polyglutamylated in contrast to the mitotic spindle. Scale bar: 4 μm. Below are shown 2 examples of representative subpellicular microtubules (dotted white box regions) stained for α/β tubulin (magenta, Alexa 568) and PolyE (green, Alexa 488). Scale bar: 5 μm. (**H**) Representative epifluorescence images of expanded *P. berghei* schizonts presenting subpellicular microtubules stained for α/β tubulin (magenta, Alexa 568) and PolyE (green, Alexa 488). Below are shown 2 examples of representative subpellicular microtubules (dotted white box regions). Scale bar: 5 μm. (**I**) Plot profile along a subpellicular microtubule displaying tubulin and PolyE signals with a schematic representation of a subpellicular microtubule on top. Magenta: tubulin, green: PolyE. Note that PolyE is not uniformly distributed along the subpellicular microtubules. (**J**) Dot plots representing the length of subpellicular microtubules in *P. falciparum* and *P. berghei*. Average length +/− standard deviation: 672 +/− 194 nm, $n = 48$ and 697 +/− 160 nm, $n = 48$ and 51, respectively. From 3 independent experiments. (**K**) Number of subpellicular microtubules in *P. falciparum* and *P. berghei*. Averages are 7.7 +/− 0.9 ($n = 45$) and 1.3 +/− 0.5 ($n = 68$), respectively. The raw data for C, F, I, J, and K can be found in S1 Data. U-ExM, ultrastructure expansion microscopy.

In *P. falciparum* segmenter schizonts, U-ExM resolved 1 to 2 subpellicular microtubules per merozoite that were on average 0.7 μm long (**Fig 3G, 3J and 3K**), as previously described by EM [4,5] or STED microscopy[13]. As opposed to the hemispindles, subpellicular microtubules exhibited a polar distribution of PolyE (**Fig 3G–3I**). Expansion of *P. berghei* segmenter schizonts revealed a similar pattern where PolyE is observed towards the apical end of the parasite. However, intracellular or free *P. berghei* merozoites contained up to 9 subpellicular microtubules (**Fig 3H and 3K**). To ascertain that the observed structures corresponded to subpellicular microtubules, we labelled proteins of expanded parasites in bulk with N-hydroxysuccinimide (NHS) ester dye conjugates, which bind to amines (**S4 Fig**). NHS-ester staining delineated the merozoite shape as well as contrasted structures with different protein density that likely correspond to the APR and the 2 secretory vesicles named rhoptries. As expected, the microtubules appeared to be located underneath the pellicle and nucleating from the APR, further confirming that they correspond to subpellicular microtubules. These differences between *P. falciparum* and *P. berghei* schizonts highlight the diversity of the apical complex organisation across species within the *Plasmodium* genus.

## The *P. berghei* ookinete possesses a remnant conoid

The apical complex of *P. falciparum* merozoites was shown to contain 3 electron dense concentric rings (referred to as polar rings [5]), with the smallest directly adjacent to the apical tip [4,5]. The third proximal ring, the APR, is much thicker than the 2 other apical rings and serves as a MTOC for the subpellicular microtubules. Despite previous observations of the apical complex by ultrastructural studies, there is relatively limited knowledge of its molecular composition and variation across the different zoite stages in *Plasmodium* [29]. We thus analysed expanded *P. berghei* ookinetes similarly stained for α/β tubulin and polyglutamylated tubulin (**Fig 4A**). A significantly larger number of subpellicular microtubules (>40) that radiate from the APR and covering most of the ookinete body were distinctly observed in expanded ookinetes, a number consistent with previous EM observations ranging from 47 to 59 [30–33]. As in segmented schizonts, the subpellicular microtubules were polyglutamylated (**Fig 4A, S5 Fig**). A less intense signal for PolyE was observed at the apical extremity of the

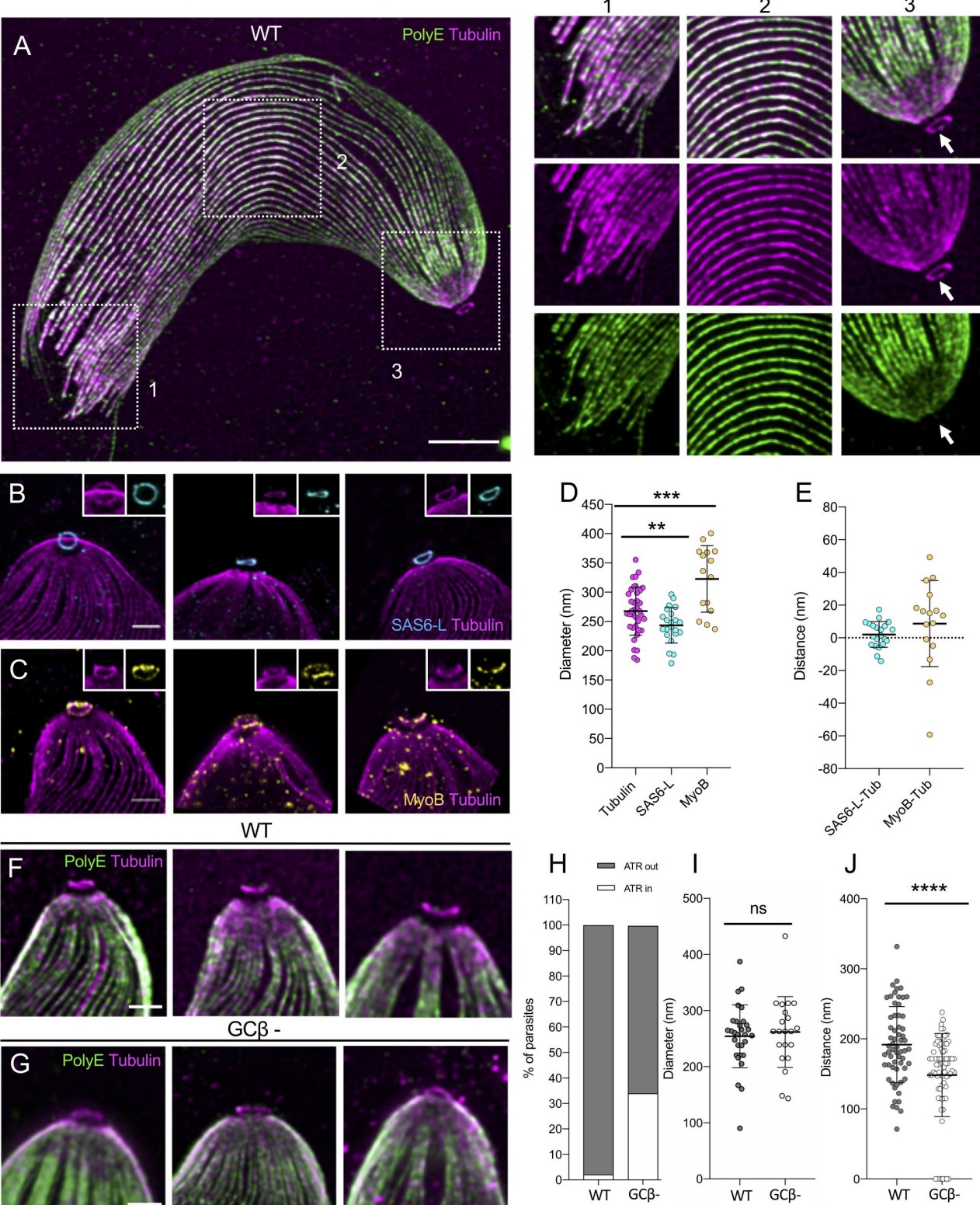

**Fig 4. Identification and characterisation of a conoid-like structure in *P. berghei* ookinetes.** (**A**) Representative confocal image of expanded ookinete stained for α/β-tubulin (magenta, Alexa 568) and PolyE (green, Alexa 488) highlighting 3 subregions boxed in white. 1: distal, 2: centre and 3: apical regions, respectively. Note that the tubulin ring or ATR in 3 is not polyglutamylated. Scale bar: 5 μm. (**B**) Gallery of the apical region of expanded ookinetes stained for α/β-tubulin (magenta, Alexa 568) and SAS6L (cyan, Alexa 488). Scale bar: 1 μm. (**C**) Gallery of the apical region of expanded

ookinetes stained for α- and β-tubulin (magenta, Alexa 568) and MyoB (yellow, Alexa 488). Scale bar: 1 μm. (**D**) Measure of the diameter of the ATR (Tubulin), SAS6L (cyan), and MyoB (yellow) rings. Averages and standard deviations in nm are as follows: Tubulin: 268 +/− 41 (*n* = 37), SAS6L: 243 +/− 30 (*n* = 22), MyoB: 323 +/− 57 (*n* = 15). Data from 2 independent experiments. Student *t* test: *p* = 0.0186 (SAS6L versus Tubulin) and *p* = 0.0003 (MyoB versus Tubulin). (**E**) Distance between the tubulin rings and SAS6L (cyan) or MyoB (yellow). Averages and standard deviations in nm are as follows: SAS6L-Tub: 2 +/− 8 (*n* = 21) and MyoB-Tub: 9 +/− 26 (*n* = 16). Data from 2 independent experiments. (**F, G**) Gallery of the apical region of expanded WT (F) and GCβ⁻ mutant (G) ookinetes stained for α/β-tubulin (magenta, Alexa 568) and PolyE (green, Alexa 488). Scale bar: 1 μm. (**H**) Percentage of WT (*n* = 56) and GCβ⁻ mutant ookinetes (*n* = 47) displaying a visible or collapsed ATR from 3 independent experiments. (**I**) Measurement of the ATR diameter in WT and GCβ⁻ expanded ookinetes. Averages and standard deviations in nm are as follows: WT = 254 +/− 56 (*n* = 30) and GCβ⁻ = 262 +/− 63 (*n* = 21). From 3 independent experiments. We observed no difference in the measured diameters in the 2 conditions. Student *t* test: *p* = 0.6534. ns = not significant. (**J**) Distance between the apical pole to the ATR in WT and GCβ⁻ mutant expanded ookinetes. Averages and standard deviations in nm are as follows: WT = 192 +/− 54 (*n* = 61) and GCβ⁻ = 145 +/− 63 (*n* = 79). From 3 independent experiments. Student *t* test: *p* < 0.0001. The raw data for D, E, H, I, and J can be found in S1 Data. ATR, apical tubulin ring; GCβ, guanylyl cyclase beta; SAS6L, SAS-6-like; WT, wild-type.

ookinete, likely corresponding to the electron dense collar structure region [34] (**Fig 4A**, **S5 Fig**).

Importantly, the improved resolution of U-ExM in ookinetes allowed us to observe a ring of tubulin above the APR that we named the ATR (**Fig 4A**). We then separately stained for α and β tubulin and found that both are present at the ATR, suggesting that it is composed by α-β heterodimers (**S6 Fig**). The observation of the ATR was unexpected as such a ring of tubulin is reminiscent of a conoid found in coccidian but thought to be lost in the Haemosporidia order containing *Plasmodium* [2]. In *T. gondii* tachyzoites, the conoid is a central component of the apical complex and consists in a set of tubulin fibres that form a cone-shaped structure located between the APR and 2 additional apical rings called the preconoidal rings [35]. A pair of intra-conoidal microtubules additionally traverses the preconoidal rings and the conoid. We thus wondered whether the ATR could represent a structure related to the conoid in *Plasmodium*.

First, we noticed that the ATR is not polyglutamylated consistent with previous observations for the conoid of *T. gondii* [16] (**Fig 4A**). Of relevance, some proteins associated with the conoid in *T. gondii* [29, 35] are also conserved in *Plasmodium* [29]. This includes the SAS-6-like (SAS6L) protein that has been demonstrated to form an apical ring in *Plasmodium* ookinetes [36]. In expanded ookinetes, endogenously tagged SAS6L- green fluorescent protein (GFP) [36] colocalised with the ATR (**Fig 4B**). While the distance between the SAS6L-GFP/ATR ring and the APR was the same, the diameter of the SAS6L-GFP ring was slightly shorter suggesting that SAS6L-GFP or at least its carboxyl terminus is facing the interior of the ATR (**Fig 4D and 4E**). In *Plasmodium*, MyoB was also shown to localise at a discrete apical location in merozoite and ookinete [37] and was suggested to fulfil, at least in part, a similar role as the conoid-associated myosin H in *T. gondii* [38]. U-ExM revealed that endogenously tagged MyoB-GFP [37] formed a dotty ring at the ookinete apex (**Fig 4C**). However, this ring was larger and slightly above the ATR/SAS6L-GFP ring, further suggesting the presence of multiple apical rings in ookinetes (**Fig 4C–4E**). However, despite the presence of the ATR, no intra-conoidal microtubule could be observed.

Another characteristic of the *T. gondii* conoid lies in its ability to protrude through the APR upon stimulation of egress and motility [9]. We thus tested whether the ATR position was linked to the motility of ookinete by comparing its position in motile and nonmotile ookinetes (**Fig 4F–4J**). Ookinetes rely on the cGMP-dependent protein kinase G (PKG) to sustain productive secretion and gliding motility [39,40]. To determine the position of the ATR upon PKG activation in wild-type parasites, we imaged mutants of the cGMP-producing guanylyl cyclase beta (GCβ) that show severely reduced motility, with only rare bouts of slow gliding [39,41,42]. In motile wild-type ookinetes, the distance between the APR and the ATR was of 200 nm, while in 34% of GCβ⁻ ookinetes, the ATR was in close proximity to the APR (**Fig**

**4H**). The remaining mutant parasites showed a significantly reduced distance of 145 nm between the ATR and the APR (**Fig 4J**, **S7 Fig**). These results suggest that the position of the ATR relative to the APR depends on the activation of motility in ookinetes.

To determine the position of both the APR and the ATR in a morphological spatial context, we labelled proteins of expanded ookinetes in bulk with NHS ester dye conjugates. NHS-ester staining delineated the ookinete shape as well as contrasted regions with different protein density that likely correspond to the collar, the apical termini of the inner membrane complex (IMC, also termed alveoli) and the subpellicular microtubules (**Fig 5A**, **S8 Fig**). Further staining for α/β tubulin revealed that the APR is located within the third upper end of the collar, while the ATR is likely in line with the IMC and the collar as inferred by EM (**Fig 5C–5J**, **S1 Video**). An additional protrusion was also observed above the ATR by NHS-ester staining, which may correspond to a small protrusion of the plasma membrane frequently observed by EM (**Fig 5C–5F**).

## Discussion

Our results demonstrate that U-ExM can be applied successfully to several *Plasmodium* species and life cycle stages to unveil ultrastructural details of the microtubule cytoskeleton that was, up to now, only reachable by EM. Imaging axoneme formation during microgametogenesis gave new insights into their mode of assembly (**Fig 6A**). For example, we show that posttranslational modifications such as PolyE occur in forming axonemes. We also observed different and nonexclusive states with either less than 8 axonemes per microgametocyte, axonemes with different lengths within a cell, or frequent misincorporation of microtubules towards the distal extremity of the axoneme. It remains unknown whether these observations reflect transient intermediate stages of assembly or terminal errors, as previously suggested [23]. However, as free gametes did not show obvious defects in axoneme assembly, it is tempting to speculate that microtubules may still grow and bundle upon the initiation of beating. Another possibility would be that only fully formed mature axonemes initiate efficient beating to form motile microgametes even in the presence of other defective axonemes in the same microgametocyte (**S2 Fig**). Further experimental characterisations will be required to test these hypotheses.

The most striking feature revealed by U-ExM was the presence of an ATR in ookinetes that most likely corresponds to a remnant conoid. The structure of the ATR appears significantly reduced or compacted compared with the spiralling tubulin fibres that form the conoid in other members of Apicomplexa. It remains unknown whether the ATR is of the same nature as the tubulin fibres forming the conoid that adopt a curvature that is not compatible with microtubules [6]. Altogether, multiple lines of evidence indicate that the ATR is related to the conoid (**Fig 6B**). First, the ATR colocalises closely with the SAS6L ring [36] that is associated with the conoid in *T. gondii* [29,35]. Second, the ATR position relative to the APR depends on the activation of ookinete motility and secretion, a characteristic reminiscent of the enigmatic protrusion of the conoid that happens upon activation of secretion and motility in *T. gondii* tachyzoites [9]. Third, the ATR is associated with at least 1 additional tightly apposed ring that may correspond to one of the so-called preconoidal rings in *T. gondii*. This additional ring is composed of MyoB, which was proposed to fulfil, at least in part, a similar role as the conoid-associated myosin H in *T. gondii* [38]. Interestingly, while MyoB is also expressed in schizonts [37], the ATR/SAS6L ring is not observed at this stage [36], further confirming that the *Plasmodium* apical complex shows stage-specific variations in its composition, as previously proposed [29,36]. In line with this, our results also reveal additional species specificities at given stages with the observation of different numbers of subpellicular microtubules between *P. berghei* and *P. falciparum* merozoites.

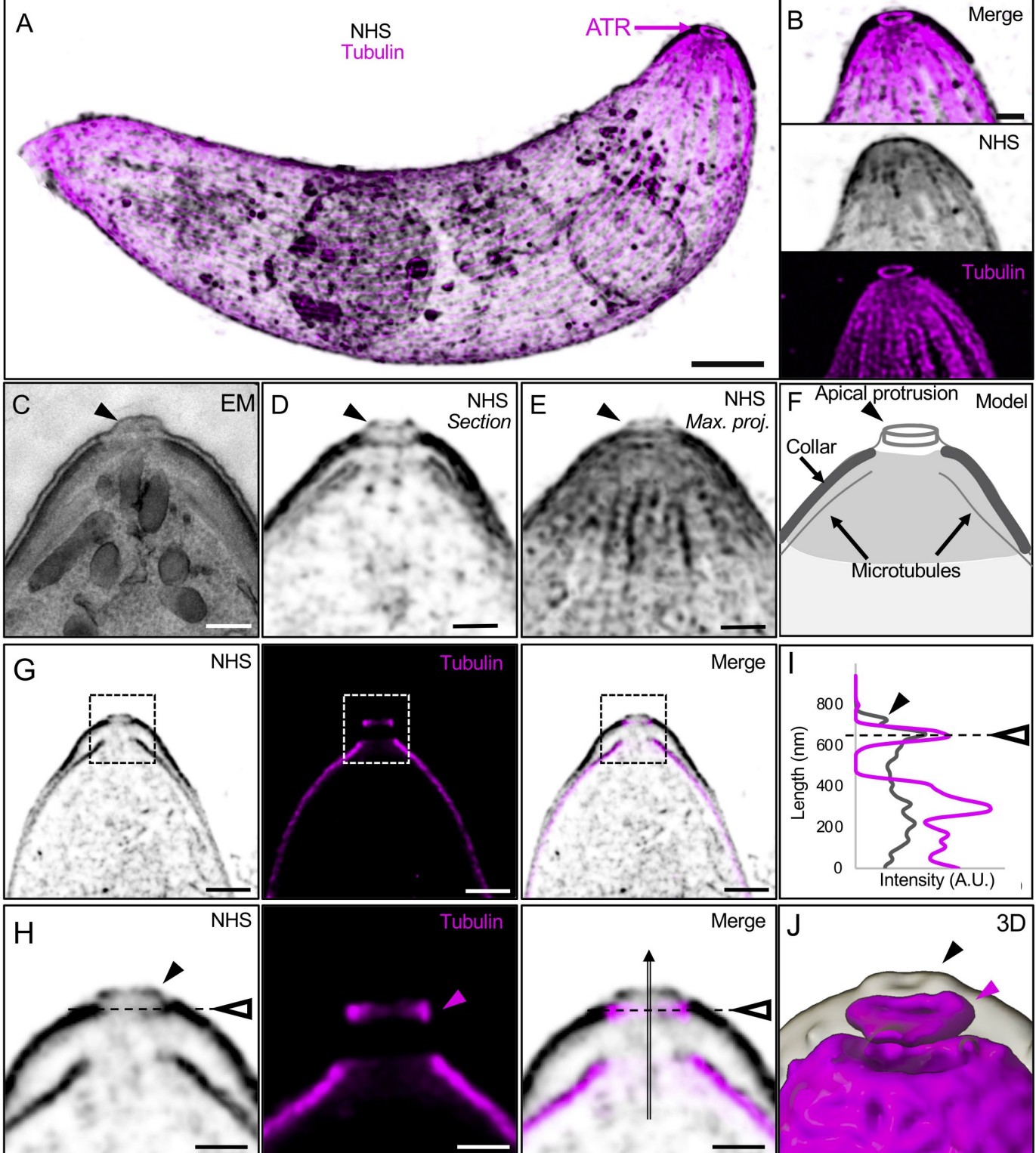

**Fig 5. U-ExM coupled with NHS-ester/Tubulin staining reveals the position of the ATR.** (**A**) Merged representative confocal image of an expanded ookinete stained for α/β-tubulin (magenta) and NHS-ester (grey). The magenta arrow indicates the position of the ATR. Scale bar: 1 μm. (**B**) Insets of the apical region from (A) highlighting the position of the ATR relative to the general ookinete structural features. (**C**) EM image of the apical region of an ookinete. Black arrowhead points to the apical protrusion. Scale bar: 250 nm. (**D, E**) Section (D) or maximum intensity projection (E) of the apical region of an expanded ookinete stained for NHS-ester from an entire image stack. Black arrowhead points to the apical protrusion. Scale bar: 250 nm. (**F**) Schematic representation of

the apical region highlighting the apical protrusion, the collar, and subpellicular microtubules. (**G**) Section of a stack of an expanded ookinete stained with NHS-ester (grey) and α/β-tubulin (magenta). Dotted square indicates the position of the zoom shown in H. Scale bar: 500 nm. (**H**) Zoom in from image in G. Dotted line with the open arrowhead indicates the position of the ATR, the black arrowhead shows the apical protrusion, the magenta arrowhead points to the ATR, and the black arrow to the position of the plot profile (I). Scale bar: 250 nm. (**I**) Plot profile intensity over length in nm showing the position of the ATR relative to the apical protrusion. Note that the ATR position is in line with the end of the collar and/or the IMC, below the apical protrusion. The underlying data can be found in S1 Data. (**J**) Three-dimensional rendering of an NHS-ester/tubulin overlay (grey and magenta, respectively) of an expanded ookinete. Black and magenta arrowheads indicate the apical protrusion and ATR, respectively. ATR, apical tubulin ring; EM, electron microscopy; IMC, inner membrane complex; NHS, N-hydroxysuccinimide; U-ExM, ultrastructure expansion microscopy.

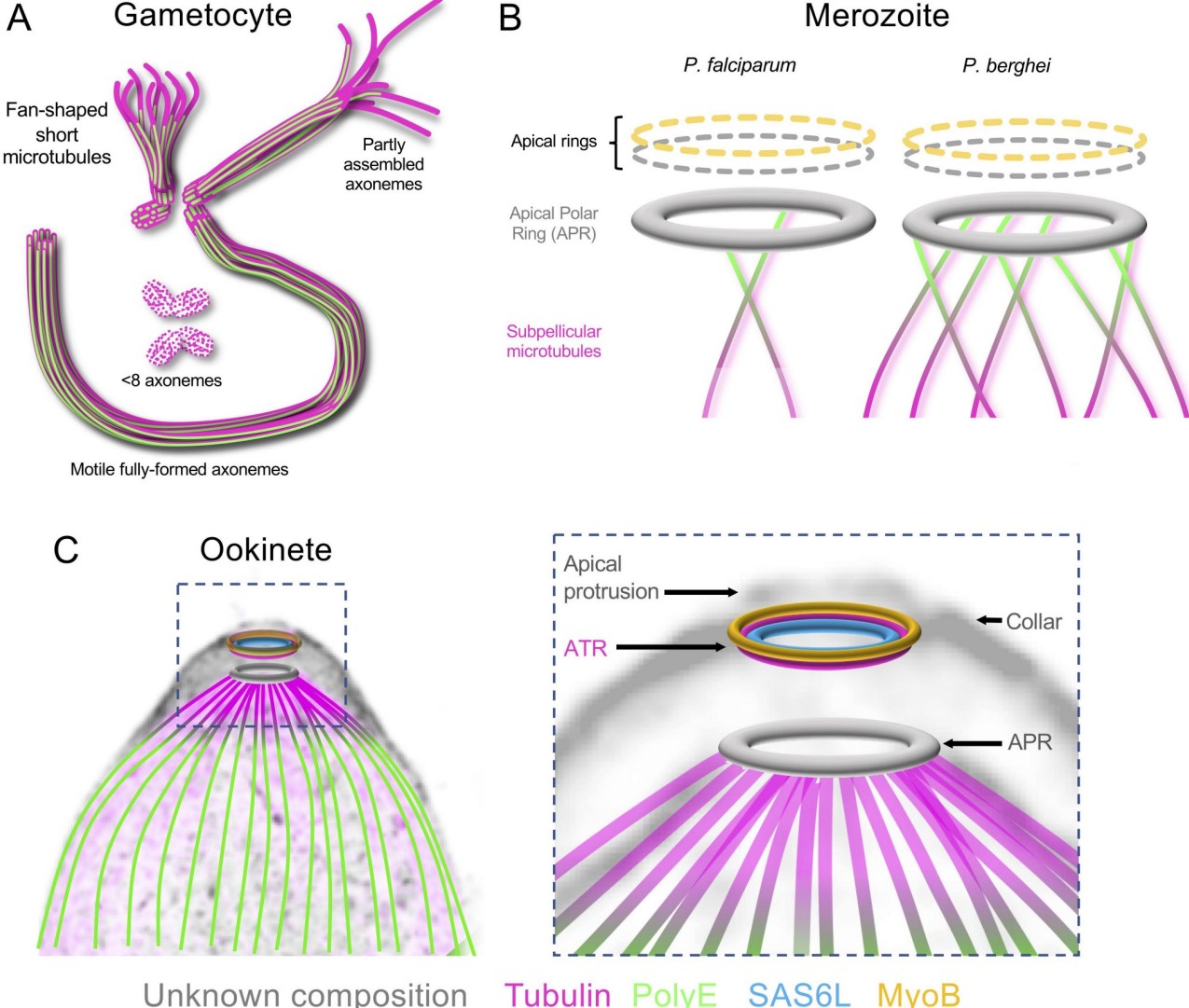

**Fig 6. Current models.** (**A**) Proposed model of axonemal assembly in microgametocytes featuring the coexistence of short fan-shaped or basket-like microtubules with partly assembled and fully assembled axonemes. Basal bodies are represented at the base of axonemes. Microtubules: magenta and PolyE: green. (**B**) Proposed models of the apical complex organisation in *P. falciparum* and *P. berghei* merozoites illustrating the diversity of the subpellicular microtubules organisation. The orientation of the polyglutamylation profile of the subpellicular microtubules in the *P. berghei* merozoite is based on **Fig 3H**. Grey denotes the presence of additional electron dense rings observed by EM whose molecular composition remains unknown. The relative position of MyoB (dotted yellow) in merozoite is currently unknown. (**C**) Model of the apical complex organisation in *P. berghei* ookinete superimposed on an NHS-ester stained expanded ookinete. This model highlights a conoid-like structure present in ookinete, composed of an ATR (magenta) and SAS6L (cyan) above the APR (grey). An additional MyoB ring is also detected in ookinete (yellow). The complex made of the ATR and SAS6L/MyoB rings is in line with the collar and/or the IMC, below the apical protrusion and above the APR. APR, apical polar ring; ATR, apical tubulin ring; IMC, inner membrane complex; SAS6L, SAS-6-like.

While the ATR likely corresponds to a remnant conoid, its position in line with the IMC and the collar in ookinete is remarkable. This suggests that when the ATR moves away from the APR, the whole apex of the ookinete does so, including the IMC and the collar. It thus seems that the apical complex is internalised within the *Plasmodium* ookinete. As in *T. gondii* tachyzoites, no obvious structure between the APR and the ATR can be highlighted by NHS-ester staining or EM, and it remains unknown how the relative position between the ATR and the APR is maintained. We also observed that a small protrusion of the plasma membrane frequently observed by EM is above the ATR. It is possible that additional preconoidal ring proteins create this extension but that the associated structures are not as electron dense as known preconoidal rings in *T. gondii* tachyzoites or *P. falciparum* merozoites to be assigned as such by EM or NHS-ester U-ExM. In addition to SAS6L and MyoB, several other proteins associated with the conoid in *T. gondii* have been recently shown to be expressed in *Plasmodium* schizonts, ookinetes, and sporozoites [9,29] and may localise at different locations in the apical complex. For example, the recently described PBANKA_1334800 that shows radial spines at the apical end of ookinetes [29] is likely one of the proteins in between the IMC and the apical ends of the subpellicular microtubules. We expect that U-ExM will allow defining the exact position of apical complex proteins in multiple *Plasmodium* stages.

We thus propose that *Plasmodium* parasites retained a conoid that is significantly divergent and reduced compared to the best described *T. gondii* conoid, which likely prevented its identification. Our results also suggest that the tubulin structure that initially defined the conoid is not expressed in all *Plasmodium* zoites, raising new questions on the molecular nature of the conoid. It now appears that the conoid is likely a conserved organelle in *Plasmodium*. However, its molecular composition is more diverse than initially expected, and the physiological requirements associated with this diversity remain mysterious. It was suggested that the conoid plays a mechanical role in the invasion by enteric apicomplexan parasites that must penetrate the thick barrier of the intestinal epithelium of vertebrates [2]. The observed variations in the molecular composition of the conoid between schizont and ookinete or the different numbers of subpellicular microtubules between *P. berghei* and *P. falciparum* schizonts thus likely reflect adaptations of the apical complex to the different host cells or environment encountered by these zoites.

Here, expansion microscopy reveals new features of the *Plasmodium* apical complex that have not been characterised before. Moreover, expansion microscopy methods have been successfully coupled with super-resolution approaches, leading to a further increase in resolution [14]. Such approaches might pave the way for further investigation of the *Plasmodium* cytoskeleton and its molecular organisation. Finally, the recently developed membrane-based expansion microscopy protocols [43, 44] could also provide other exciting insights into the membrane biology of *Plasmodium*. In summary, the use of expansion microscopy can nicely complement electron and super-resolution microscopy as it provides molecular details to the available structural information.

## Material and methods

All primary and secondary antibodies used in this study can be found in **S1 Table**.

### Ethics statement

All animal experiments were conducted with the authorisation numbers GE/82/15 and GE/41/17, according to the guidelines and regulations issued by the Swiss Federal Veterinary Office.

## *P. berghei* maintenance and production

*P. berghei* ANKA strain-derived [45] clone 2.34 [46] together with derived transgenic lines Kin8B-KO [24], SAS6L-GFP [36], and MyoB-GFP [37] were grown and maintained in CD1 outbred mice as previously described [47]. Moreover, 6- to 10-week-old mice were obtained from Charles River Laboratories (France), and females were used for all experiments. Mice were specific pathogen free (including *Mycoplasma pulmonis*) and subjected to regular pathogen monitoring by sentinel screening. They were housed in individually ventilated cages furnished with a cardboard mouse house and Nestlet, maintained at 21 ± 2˚C under a 12-hour light/dark cycle, and given commercially prepared autoclaved dry rodent diet and water ad libitum. The parasitaemia of infected animals was determined by microscopy of methanol-fixed and Giemsa-stained thin blood smears.

Schizonts were purified from overnight in vitro culture (RPMI1640 with 25 mM glutamine, HEPES + 10 mM NaHCO$_3$, 20% foetal bovine serum and 100 U/ml penicillin, 100 μg/ml streptomycin) on a Histodenz cushion made from 55% of the Histodenz stock and 45% PBS. Parasites were harvested from the interface and collected by centrifugation at 500 g for 3 minutes. Gametocyte production and purification was performed as previously described [47]. Parasites were grown in mice that had been phenyl hydrazine-treated 3 days before infection. One day after infection, sulfadiazine (20 mg/L) was added in the drinking water to eliminate asexually replicating parasites. For gametocyte purification, parasites were harvested in suspended animation medium (SA; RPMI 1640 containing 25 mM HEPES, 5% fetal calf serum [FCS], 4 mM sodium bicarbonate, pH 7.20) and separated from uninfected erythrocytes on a Histodenz cushion made from 48% of a Histodenz stock (27.6% [w/v] Histodenz [Sigma/Alere Technologies, Germany] in 5.0 mM TrisHCl, 3.0 mM KCl, 0.3 mM EDTA, pH 7.20) and 52% SA, final pH 7.2. Gametocytes were harvested from the interface. Gametocytes were activated in RPMI 1640 containing 25 mM HEPES, 4 mM sodium bicarbonate, 5% FCS, 100 μM xanthurenic acid, pH 7.4).

For western blot analyses, gametocytes were lysed in RIPA buffer (50 mM Tris HCl pH 8, 150 mM NaCl, 1% NP-40, 0.5% sodium deoxycholate, 0.1% SDS), supplemented with 1 × protease inhibitor cocktail (Thermo Fisher Scientific, Life Technologies Europe BV) on ice for 30 minutes. Proteins were detected using the primary antibodies α-tubulin (1:1,000) and α-PolyE (1:2,000) and secondary antibodies α-mouse (1:3,000) and α-rabbit (1:3,000). Ookinete cultures were performed as previously described [40]. Parasites were maintained in phenyl hydrazine-treated mice. Ookinetes were produced in vitro by adding 1 volume of high gametocytaemia blood in 30 volumes of ookinete medium (RPMI1640 containing 25 mM HEPES, 10% FCS, 100 μM xanthurenic acid, pH 7.5) and incubated at 19˚C for 18 to 24 hours. Ookinetes were purified using paramagnetic anti-mouse IgG beads (Life Technologies Europe BV) coated with anti-p28 mouse monoclonal antibody (13.1).

## *P. falciparum* culture

*P. falciparum* strain 3D7, a clone from the NF54 isolate [48], was grown in human erythrocytes in RPMI-1640 medium with glutamine (Gibco, Life Technologies Europe BV), 0.2% sodium bicarbonate, 25 mM HEPES, 0.2% glucose, 5% human serum, and 0.1% Albumax II (Life Technologies Europe BV). Parasite cultures were kept synchronised by double sorbitol treatments. Late stage parasites were purified from synchronous cultures using Percoll (GE Healthcare, Sigma, Germany).

## U-ExM

Ookinetes and schizonts were centrifuged at 1,000 rpm during 5 minutes at 24˚C and resuspended in 500 μL of PBS 1X. Parasites were sedimented on poly-D-lysine (A-003-E, Sigma)

coverslips (150 μL/coverslip) during 10 minutes at room temperature (RT). To stop the activation of the gametocytes, the same protocol was used but keeping the parasites at 4˚C. Then parasites were fixed in −20˚C methanol during 7 minutes and prepared for U-ExM as previously published [14]. Briefly, coverslips were incubated for 5 hours in 2X 1.4% AA/ 2% FA mix at 37˚C prior gelation in APS/Temed/Monomer solution (19% Sodium Acrylate; 10% AA; 0,1% BIS-AA in PBS 10X) during 1 hour at 37˚C. Then denaturation was performed during 1 hour and 30 minutes at 95˚C [49]. After denaturation, gels were incubated in ddH2O at RT during 30 minutes. Next, gels were incubated in ddH2O overnight for complete expansion. The following day, gels were washed in PBS twice for 15 minutes to remove excess water before incubation with primary antibody solution. They were stained 3 hours at 37˚C with primary antibodies against PolyE (1:500), α-tubulin and β-tubulin (1:200), centrin (1:300), anti-GFP (1:250). Gel were washed 3 × 10 minutes in PBS-Tween 0.1% prior incubation with secondary antibodies (anti-mouse Alexa 568, Anti-mouse 488, anti-rabbit Alexa 488, Anti-guinea pig Alexa 568–1:400) during 3 hours at 37˚C and 3 washes of 10 minutes in PBS-Tween. Overnight, a second round of expansion was done in water before imaging.

## NHS-ester staining

Directly after antibody staining, the gel where incubated in NHS Ester (Thermo Fisher Scientific catalog number: 46402) and diluted at 10 μg/mL in PBS for 1 hour and 30 minutes at RT on a rocking platform. The gels where then washed 3 times for 15 minutes with PBS-Tween 0.1% then expanded overnight in ultrapure water.

Imaging was performed on a Leica Thunder inverted microscope (Germany) using 63X 1.4NA oil objective with Small Volume Computational Clearing mode to obtain deconvolved images. Three-dimensional stacks were acquired with 0.21 μm z-interval and x,y pixel size of 105 nm. Images were analysed and merged using ImageJ software.

Confocal microscopy was performed on a Leica TCS SP8 (Germany) with a 63×/1.4-NA (numerical aperture) oil immersion objective, using the HyVolution mode20 to generate deconvolved images, with the following parameters: "HyVolution Grade" at max resolution, Huygens Essential as "Approach," water as "Mounting Medium," and Best Resolution as "Strategy."

## Measurements

The diameter and distance between the ring of tubulin and the microtubules were always measured from dual staining experiments. After applying a maximum projection of the stack, we used the homemade plug-in pickCentrioleDim on Fiji software developed by M. Le Guennec to measure the peak-to-peak distance for each fluorescent channel for each parasite. The expansion factor of each gel was after applied to the value to obtain the real distance.

To measure the length of microtubules, we used Fiji to draw a line to define our region of interest (ROI) following tubulin staining and determined its associated length. The expansion factor of each gel was after applied to the value to obtain the real size.

## Transmission electron microscopy

EM was performed as previously described [40]. RBCs infected with *P. berghei* extracellular ookinetes were fixed with 2.5% glutaraldehyde (Electron Microscopy Sciences, United States of America) and 2.0% paraformaldehyde (Electron Microscopy Sciences) in 10 mM PBS pH 7.4 for 1 hour at room temperature. Pelleted cells were embedded in 3% low-melted agarose (Eurobio, France) in 10 mM PBS and dissected in small pieces in order of easier handling and to prevent loss of cells during subsequent processing steps. After extensive washing (5 × 5

minutes) in 0.1 M sodium cacodylate buffer pH7.4 (Sigma), samples were postfixed with 1% osmium tetroxide (Electron Microscopy Sciences) reduced with 1.5% ferrocyanide (Sigma) in 0.1 M sodium cacodylate buffer pH 7.4 for 1 hour at room temperature, followed by postfixation with 1% osmium tetroxide alone (Electron Microscopy Sciences) in 0.1 M sodium cacodylate buffer pH 7.4 for 1 hour at room temperature. After washing with double distilled water (2 × 5 minutes) samples were en-block post-stained with 1% aqueous uranyl acetate (Electron Microscopy Sciences) for 1 hour at room temperature. Samples were then washed with double distilled water for 5 minutes and dehydrated in graded ethanol series (2 × 50%, 1 × 70%, 1 × 90%, 1 × 95%, and 2 × 100% for 3 minutes each wash). Samples were then infiltrated at room temperature with Durcupan resin (Sigma) mixed 100% ethanol at 1:2, 1:1, and 2:1 for 30 minutes each step, followed by fresh pure Durcupan resin for 2 × 30 minutes and transferred into fresh pure Durcupan resin for 2 hours. Finally, samples were embedded in fresh Durcupan resin filled small thin-wall PCR tubes and polymerised and cured at 60˚C for 24 hours. Ultrathin sections (60 nm) were cut with Leica Ultracut UCT microtome (Leica Microsystems, Germany) and diamond knife (DiATOME) and collected onto 2 mm single slot copper grids (Electron Microscopy Sciences) coated with 1% Pioloform plastic support film. Sections were then examined and transmission electron microscopy (TEM) images collected using Tecnai 20 TEM (FEI) electron microscope operating at 80 kV and equipped with a side-mounted Mega-View III CCD camera (Olympus Soft Imaging Systems, Germany) controlled by iTEM acquisition software (Olympus Soft Imaging Systems).

## Supporting information

**S1 Data. Raw data and statistical analyses for Figs 1H, 3C, 3F, 3I–3K, 4D, 4E, 4H–4J and 5I and S1H, S3B–S3E and S4E Figs.**
(XLSX)

**S2 Data. Original scans of the western blot shown in S3 Fig.**
(TIF)

**S1 Fig. Measurements to assess the isotropicity of U-ExM expanded specimens.** Measurement skills to evaluate parasite size before and after expansion. Dotted lines represent the ROI used for the measurements for each parasite stage: microgametocytes, non-expanded (**A**), and expanded (**B**); ookinetes, non-expanded (**C**), and expanded (**D**); schizonts, non-expanded (**E**), and expanded (**F**). Scale bar: 5 μm. (**G**) Ratio between the size before and after expansion and the average expansion factor (gel size/coverslip size). Averages and standard deviations are given in the table from at least 3 independent experiments for each stage. Related to Fig 1 (*n* are indicated in the legend of Fig 1H). (**H**) Plot profile of the ROI extracted from 1 ookinete image stained for α/β-tubulin (magenta, Alexa 568) showing the resolution limit of U-ExM. The example of ROI used for this plot profile is indicated in green on panel D. The average distance between 2 microtubules is of 137 nm +/− 17. The corresponding raw data can be found in S1 Data. ROI, region of interest; U-ExM, ultrastructure expansion microscopy.
(TIFF)

**S2 Fig. Gallery of U-ExM expanded WT *Plasmodium* microgametocytes.** Expanded WT microgametocytes and gametes stained for α- and β-tubulin (magenta, Alexa 568) and PolyE (green, Alexa 488). Arrows show apparent differences between tubulin and PolyE staining. The white asterisk denotes fully formed axonemes. The white arrowhead indicates an exflagellated gamete. Scale bar: 5 μm. U-ExM, ultrastructure expansion microscopy; WT, wild-type.
(TIFF)

**S3 Fig. Quantification of polyglutamylated tubulin in expanded microgametocytes. (A)** Wide-field representative image of an expanded microgametocyte stained for α- and β-tubulin (magenta, Alexa 568) and PolyE (green, Alexa 488). White arrows (1, 2, 3) indicate the position and orientation of the plot profiles shown in B to D. **(B–D)** Plot profiles across axonemes and tubulin structures/MTs showing a higher level of PolyE signal in complete axonemes in comparison with tubulin structures/microtubules (MT). **(E)** Western blot analysis showing tubulin and polyglutamylated tubulin in WT and *P. berghei* Kin8B-KO gametocytes 15 minutes postactivation by XA. The histograms show the relative values from 3 independent biological replicates. The raw data for B, C, D, and E can be found in S1 Data and the original scans in S2 Data, respectively. MT, microtubule; WT, wild-type.
(TIFF)

**S4 Fig. NHS-ester staining coupled with U-ExM of *P. berghei* schizonts. (A–C)** Merge widefield image (A) of an expanded schizont stained for NHS-ester (grey) (B) and α- and β-tubulin (magenta, Alexa 568) (C). Arrowheads indicate the position of the APR seen as a grey ring. Dotted square boxes corresponds to the insets. Scale bar: 4 μm. **(D)** Confocal image of a free expanded merozoite stained with NHS-ester (grey) and α- and β-tubulin (magenta, Alexa 568). The arrowhead indicates the position of the APR seen as a grey ring. Scale bar: 2 μm. **(E)** Plot profile along the dotted line in D. Note that the measured APR have an average diameter of 183 nm +/ 12 (*n* = 13). The corresponding raw data can be found in S1 Data. APR, apical polar ring; NHS, N-hydroxysuccinimide; U-ExM, ultrastructure expansion microscopy.
(TIFF)

**S5 Fig. U-ExM of WT *Plasmodium* ookinetes. (A)** Representative confocal image of WT ookinete expanded using the U-ExM protocol and stained for α/β-tubulin (magenta, Alexa 568) and PolyE (green, Alexa 488). The 2 channels are represented independently. Scale bar: 5 μm. **(B)** Representative gallery of confocal images of WT ookinetes. Ookinetes were expanded using the U-ExM protocol and stained for α/β-tubulin (magenta, Alexa 568) and PolyE (green, Alexa 488). Scale bar: 5 μm. Insets show the delineated white-boxed region. The ATR is not polyglutamylated, while the subpellicular microtubules are PolyE positive. ATR, apical tubulin ring; U-ExM, ultrastructure expansion microscopy; WT, wild-type.
(TIFF)

**S6 Fig. The ATR is composed of both α- and β-tubulin. (A, B)** Representative wide-field images of WT ookinete expanded using the U-ExM protocol and stained for α- (A) or β- (B) tubulin (magenta, Alexa 568). Scale bar: 2 μm. ATR, apical tubulin ring; U-ExM, ultrastructure expansion microscopy; WT, wild-type.
(TIFF)

**S7 Fig. U-ExM of expanded GCβ⁻ *Plasmodium* ookinetes.** Representative confocal images of a GCβ⁻ ookinete mutant. Ookinetes were expanded using the U-ExM protocol and stained for α/β-tubulin (magenta, Alexa 568) and PolyE (green, Alexa 488). Scale bar: 5 μm. The ATR is closely associated with the APR in contrast with WT ookinetes. APR, apical polar ring; ATR, apical tubulin ring; U-ExM, ultrastructure expansion microscopy; WT, wild-type.
(TIFF)

**S8 Fig. NHS-ester staining of expanded ookinetes. (A–C)** Representative confocal image of an expanded ookinete stained for α/β-tubulin (magenta, Alexa 568) and NHS-ester (grey). Merge (A), α/β-tubulin staining (B) and NHS-staining (C). Scale bar: 5 μm. NHS-ester staining reveals the position of the collar and the nucleus. Microtubules are only weakly detected, and micronemes are not stained by NHS-ester in contrast with the collar. NHS, N-

hydroxysuccinimide.
(TIFF)

**S1 Video.** Three-dimensional rendering of an expanded ookinete stained for α/β-tubulin (magenta, Alexa 568) and NHS-ester (grey). NHS, N-hydroxysuccinimide.
(MOV)

**S1 Table. Antibodies and parasite lines used in this study.** All cell lines as well as primary and secondary antibodies used in this study are listed, including the dilution used, the source or reference, and the catalogue number.
(DOCX)

## Acknowledgments

We thank the BioImaging Center at Unige as well as Philippe Bastin, Linda Kohl, Nicolas Dos Santos Pacheco, Julien Guizetti, and Nikolai Klena for critical reading of the manuscript and insightful comments. We also would like to thank Rita Tewari for generously sharing the *P. berghei* Kin8B-KO, SAS6L-GFP, and MyoB-GFP lines.

## Author Contributions

**Conceptualization:** Mathieu Brochet, Paul Guichard, Virginie Hamel.

**Formal analysis:** Eloïse Bertiaux, Vincent Louvel, Bohumil Maco, Paul Guichard, Virginie Hamel.

**Funding acquisition:** Eloïse Bertiaux, Dominique Soldati-Favre, Mathieu Brochet, Paul Guichard.

**Methodology:** Eloïse Bertiaux, Aurélia C. Balestra, Lorène Bournonville, Vincent Louvel, Virginie Hamel.

**Resources:** Aurélia C. Balestra.

**Supervision:** Mathieu Brochet, Paul Guichard, Virginie Hamel.

**Validation:** Eloïse Bertiaux.

**Visualization:** Eloïse Bertiaux, Lorène Bournonville, Vincent Louvel, Bohumil Maco, Paul Guichard, Virginie Hamel.

**Writing – original draft:** Mathieu Brochet, Paul Guichard, Virginie Hamel.

**Writing – review & editing:** Eloïse Bertiaux, Aurélia C. Balestra, Dominique Soldati-Favre, Mathieu Brochet, Paul Guichard, Virginie Hamel.

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
