## [Editor Report · Decision Letter 0]

30 Oct 2020

Dear Dr. Hamel, 

Thank you for submitting your manuscript entitled "Expansion Microscopy provides new insights into the cytoskeleton of malaria parasites including the conservation of a conoid" for consideration as a Research Article by PLOS Biology.

Your manuscript has now been evaluated by the PLOS Biology editorial staff as well as by an academic editor with relevant expertise and I am writing to let you know that we will accept it without further review, but we will need to do some reporting and formatting checks first.

For this, we need you to complete your submission by providing the metadata that is required. To this end, please login to Editorial Manager where you will find the paper in the 'Submissions Needing Revisions' folder on your homepage. Please click 'Revise Submission' from the Action Links and complete all additional questions in the submission questionnaire.

Please, delete the part where you say this is an update article, since it is not updating a previous PLOS Biology paper and then our system will be categorizing this manuscript wrong. 

Please re-submit your manuscript within two working days, i.e. by Nov 01 2020 11:59PM.

Kind regards,

Paula 

---

Associate Editor

PLOS Biology

---

## [Editor Report · Decision Letter 1]

5 Nov 2020

Dear Dr. Hamel,

Thank you very much for submitting your manuscript "Expansion Microscopy provides new insights into the cytoskeleton of malaria parasites including the conservation of a conoid" for consideration as a Research Article by PLOS Biology. This paper was a Review Commons paper that was evaluated by the PLOS Biology editors as well as by an Academic Editor with relevant expertise. 

We're delighted to let you know that we're editorially satisfied with your manuscript. However before we can formally accept your paper and consider it "in press", we also need to ensure that your article conforms to our guidelines. A member of our team will be in touch shortly with a set of requests. As we can't proceed until these requirements are met, your swift response will help prevent delays to publication. Please also make sure to address the data and other policy-related requests noted at the end of this email.

- a cover letter that should detail your responses to any editorial requests, if applicable

*Copyediting*

*Published Peer Review History*

*Early Version*

Sincerely,

Paula

---

Associate Editor,

pjaureguionieva@plos.org,

PLOS Biology

DATA POLICY:

Regardless of the method selected, please ensure that you provide the individual numerical values that underlie the summary data displayed in the following figure panels as they are essential for readers to assess your analysis and to reproduce it: Figures 1H, 3C, 3F, 3I, 3J, 3K, 4D, 4E, 4H, 4I, 4J, 5I, Supplementary figures 1H, 3B, 3C, 3D, 3E, and 4E.

OTHER REMARKS:

Please, remove the data in the system where you say that this is an Update Article. This is only applicable for papers updating Articles published at PLOS Biology.

Please, remove any priority claims such as on page 10 "highlight for the first time". We consider that this is not necessary to highlight your work.

---

## [Editor Report · Decision Letter 2]

21 Jan 2021

Dear Dr. Hamel,

I am writing concerning your manuscript submitted to PLOS Biology, entitled “Expansion Microscopy provides new insights into the cytoskeleton of malaria parasites including the conservation of a conoid.”

We have now completed our final technical checks and have approved your submission for publication. You will shortly receive a letter of formal acceptance from the editor.

Kind regards,

PLOS Biology